# Biological oxygen demand optode analysis of coral reef-associated microbial communities exposed to algal exudates

AK Gregg[1], M Hatay[2], AF Haas[1], NL Robinett[1], K Barott[3], MJA Vermeij[4,5], KL Marhaver[4,7], P Meirelles[6], F Thompson[6] and F Rohwer[1]

[1] Department of Biology, San Diego State University, San Diego, CA, USA
[2] Department of Physics, San Diego State University, San Diego, CA, USA
[3] Scripps Institution of Oceanography, University of California at San Diego, La Jolla, CA, USA
[4] Caribbean Research and Management of Biodiversity (CARMABI), Willemstad, Curaçao
[5] Aquatic Microbiology, Institute for Biodiversity and Ecosystem Dynamics, University of Amsterdam, Amsterdam, The Netherlands
[6] Laboratory of Microbiology, Institute of Biology, Federal University of Rio de Janeiro, Rio de Janeiro, RJ, Brasil
[7] Department of Natural Sciences, University of California at Merced, Merced, CA, USA

Corresponding author
AK Gregg, agregg88@gmail.com

## ABSTRACT

Algae-derived dissolved organic matter has been hypothesized to induce mortality of reef building corals. One proposed killing mechanism is a zone of hypoxia created by rapidly growing microbes. To investigate this hypothesis, biological oxygen demand (BOD) optodes were used to quantify the change in oxygen concentrations of microbial communities following exposure to exudates generated by turf algae and crustose coralline algae (CCA). BOD optodes were embedded with microbial communities cultured from *Montastraea annularis* and *Mussismilia hispida*, and respiration was measured during exposure to turf and CCA exudates. The oxygen concentrations along the optodes were visualized with a low-cost Submersible Oxygen Optode Recorder (SOOpR) system. With this system we observed that exposure to exudates derived from turf algae stimulated higher oxygen drawdown by the coral-associated bacteria than CCA exudates or seawater controls. Furthermore, in both turf and CCA exudate treatments, all microbial communities (coral-, algae-associated and pelagic) contributed significantly to the observed oxygen drawdown. This suggests that the driving factor for elevated oxygen consumption rates is the source of exudates rather than the initially introduced microbial community. Our results demonstrate that exudates from turf algae may contribute to hypoxia-induced coral stress in two different coral genera as a result of increased biological oxygen demand of the local microbial community. Additionally, the SOOpR system developed here can be applied to measure the BOD of any culturable microbe or microbial community.

## INTRODUCTION

Over-fishing and eutrophication are associated with increasing abundances of turf algae (*McClanahan, Cokos & Sala, 2002*; *Dulvy, Freckleton & Polunin, 2004*; *Sandin et al., 2008*; *Vermeij et al., 2010*) and coral mortality worldwide (*Done, 1992*; *Harvell et al., 1999*; *Harvell et al., 2007*; *Hughes et al., 2007*). These degraded systems are also associated with increased microbial abundances (*Dinsdale et al., 2008*; *Bruce et al., 2012*) and greater microbial energy usage (*McDole et al., 2012*). Turf algae produce higher amounts of bioavailable dissolved organic carbon (DOC) than calcifying reef organisms such as coral and crustose coralline algae (CCA), and this DOC promotes microbial growth and respiration (*Wild et al., 2010*; *Haas et al., 2011*). Further, increasing amounts of bioavailable DOC (e.g., glucose, algal exudates) can lead to coral mortality (*Kline et al., 2006*), and these effects can be prevented by the addition of antibiotics, suggesting microbially-mediated mechanisms are involved (*Kline et al., 2006*; *Smith et al., 2006*). However, algal DOC can vary significantly across algal species in terms of its monosaccharide composition (*Nelson et al., 2013*). This variation can differentially affect the growth of pelagic microbes, with turf algae able to support greater abundances of pelagic coral reef microbes than other types of reef-associated algae (*Haas et al., 2011*). This variation can also be seen in freshwater systems where terrestrial DOC runoff from trees differentially fuels microbial communities (i.e., DOC derived from certain tree species supports greater bacterial production than DOC derived from other tree species, *Lennon & Pfaff, 2005*).

Various studies have demonstrated that dissolved oxygen (DO) concentrations are significantly reduced at coral-algal interfaces (*Barott et al., 2009*; *Barott et al., 2012*, but see *Wangpraseurt et al., 2012*) and on average over a diurnal cycle in the water column overlying algal beds (*Haas et al., 2010*; *Wild et al., 2010*). These hypoxic conditions could also be reversed with addition of antibiotics (*Smith et al., 2006*), or the removal of the alga (*Barott et al., 2012*). Taken together, this evidence supports the hypothesis that algal induced microbially-mediated hypoxia plays a significant role in interaction processes structuring the benthic reef environments. Based on these observations, the Disease, DOC, Algae and Microbes (DDAM) model has been proposed in which fleshy algal-derived DOC stimulates rapid microbial growth, which creates hypoxic zones and results in coral stress and suffocation (*Kuntz et al., 2005*; *Kline et al., 2006*; *Rohwer, Youle & Vosten, 2010*; *Dinsdale & Rohwer, 2011*; *Barott & Rohwer, 2012*). It is important to note, however, that in contrast to the non-calcifying turf and macroalgae, hypoxia has not been observed on borders between CCA and corals and these interactions do not appear to be harmful to most corals (*Barott et al., 2009*; *Vermeij et al., 2010*; *Barott et al., 2012*). Furthermore, coral recruits preferentially settle on some types of CCA (e.g., *Titanoderma prototypum, Hydrolithon* spp.) (*Morse et al., 1988*; *Harrington et al., 2004*; *Vermeij & Sandin, 2008*; *Price, 2010*). This indicates that, depending on the functional group, benthic algae can either benefit or hinder adjacent corals.

Planar optodes are a useful tool to investigate these spatially distinct dynamics in a coral reef system. Using a luminescent indicator they can help visualize the concentrations of a broad range of variables of interest (*Glud et al., 1996*) including oxygen, pH, carbon

dioxide, and glucose (for review, see *Borisov & Wolfbeis, 2008*). Another expanding area within optical sensors is the use of whole cell bacterial biosensors for the measurement of biological oxygen demand (BOD), which have been used to extrapolate water quality in aquatic environments (*Preininger, Klimant & Wolfbeis, 1994*; *Kwok et al., 2005*; *Jiang et al., 2006*; *Lin et al., 2006*). These sensors take advantage of microbial cells immobilized in a matrix and coupled to an oxygen transducer. Oxygen planar and micro-optodes have previously only been applied to the study of coral reefs under controlled lab conditions (*Wild et al., 2004*; *Kuhl & Polerecky, 2008*; *Haas et al., 2013a*). *Larsen et al. (2011)* showed for the first time that spatial oxygen dynamics can be accurately monitored using commercially available compact digital cameras. Expanding upon this method described in *Larsen et al. (2011)*, we have developed a fully submersible, timelapsed oxygen optode system (SOOpR). This system is capable of measuring oxygen concentrations in two dimensions, as well as BOD of reef-associated microbes.

Given the differences in DOC production rates (*Haas et al., 2011*), secondary metabolites (*Morrow et al., 2011*), and microbes (*Barott et al., 2011*; *Barott et al., 2012*) associated with different functional groups of benthic reef algae, we hypothesized that coral-associated microbes will exhibit differential responses to exudates from the various algal guilds. To investigate this hypothesis, we utilized BOD optodes embedded with a microbial community cultured from the corals *Montastraea annularis* and *Mussismilia hispida* to measure respiration when exposed to exudates generated by turf algae and CCA.

## MATERIALS AND METHODS

### Lighting and camera system

Images of BOD optode plates were taken every 5 min over a 7 h period using the submersible oxygen optode recorder system. SOOpR consists of an automated camera and excitation source that can be fully submerged underwater. A Powershot G11 camera (Canon, USA) was equipped with the Canon Hack Development Kit (CHDK, Firmware version 1.00 L). CHDK is an open source program that adds new features to the camera's standard options, such as timelapse photography, which was used in this study. The CHDK hack was outfitted with the Ultra Intervalometer script (originally written by *Keoeeit (2010)*; Fig. S1) to allow for timelapse photography. The camera settings were set to: shutter speed 1/20″, F-Stop 4.5, ISO 100.

The camera was encased in a WP-DC34 underwater housing (Canon, USA). Excitation light was supplied from custom built underwater strobes optically triggered by the flash of the camera. Two YS-300 strobe housings were gutted of original contents to allow space for custom lighting (Sea&Sea SUNPAK Co., USA). Strobes were equipped with three 447.5 nm Royal Blue tri-star 2730 mW light emitting diodes (LED; Phillips-Luxeon, Canada) to excite the indicator. LED were covered by a blue dichroic filter (UQG Optics, United Kingdom) to block the tail end of the light spectrum emitted. The LED were controlled with an arduino-based microcontroller, Camera Axe 5, which allows for triggering of a camera or flash using various signals (Camera Axe, USA). The Camera Axe 5 was used to trigger the LED via the optical signal of the camera's flash. The flash was coupled to a

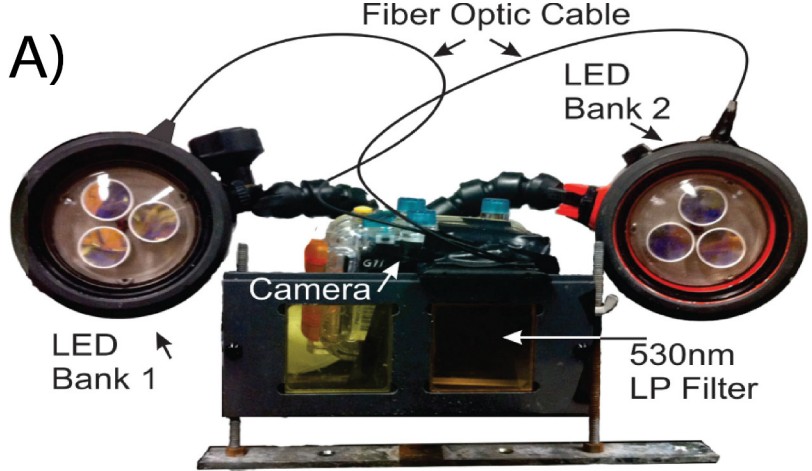

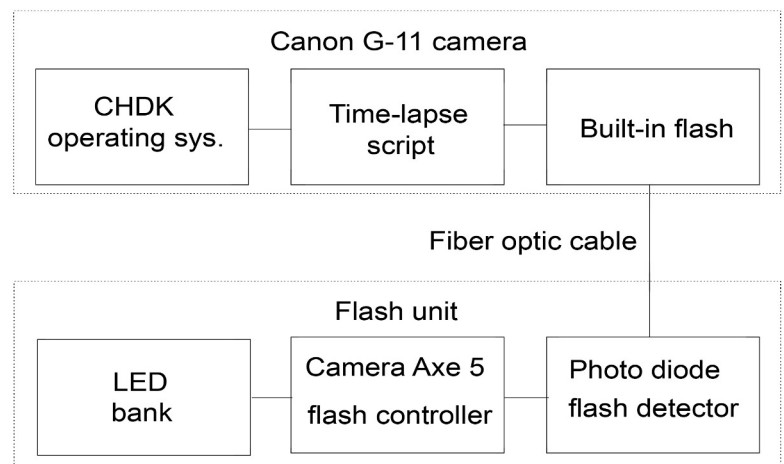

**Figure 1 Schematic of the SOOpR system.** (A) Outer components of SOOpR system: G11 camera in an underwater housing equipped with a 530 nm longpass filter. The flash from the camera is coupled to an LED bank via a fiber optic cable. Underwater strobes house three tri-star LEDs covered by 470 nm shortpass filters. (B) Inner components used for optical triggering: camera flash fiber-optically coupled to a photo detector that converts the light signal from the flash to an electrical signal and triggers the Camera Axe. The Camera Axe then outputs a signal to turn on the LED bank, allowing for excitation of the system.

photodetector (OSRAM Opto Semiconductors, USA) inside of the strobe housing using a 2000 μm fiber optic cable (Edmund Optics, USA), which signaled the Camera Axe to turn on the bank of LED for excitation (Fig. 1B). The Camera Axe settings were: bulb time of 2 s, optical trigger threshold of 25. The underwater housing was equipped with a Schott 530 nm longpass filter (UQG Optics, United Kingdom) to block the excitation source from contaminating luminescent signal. The camera system was positioned 20 cm from the subject and strobes were oriented approximately 20° from the horizontal plane of the camera (Fig. 1A).

## Construction of BOD optode plates

The carrier from a 100 μl micropipette tip box was used to construct a 96-well BOD optode plate. Spray mount adhesive (3M, USA) was applied onto one side of the carrier and a planar oxygen optode was fixed to it. The optode was oriented with the oxygen reactive side of the optode facing the interior of the wells. Care was taken to ensure that the optode exposed to the inside of the well was not disturbed by adhesive. The oxygen optode sheets were prepared according to *Larsen et al. (2011)*. The wells on the perimeter of the plate were not used.

## Calibration of optodes

A calibration was performed by recording images of each optode in known oxygen concentrations ranging from 100% air saturation to anoxia. The starting concentration of the calibration seawater was ∼250 μM DO, which was then saturated with nitrogen gas (AirGas, USA) to decrease the concentration of oxygen in the calibration chamber. Oxygen concentrations were measured using an LBOD101 Luminescent DO probe (Hach Lange, Germany), and images were taken at approximately 30 μM step decreases until water reached anoxia. The ratios of these images were then used to determine $K_{sv}$ and $\alpha$ for each plate (discussed below).

## Experiment 1 - Algal exudate studies

*Culturing microbes: Montastraea annularis* and *Mussismilia hispida* mucus samples were collected at ∼5 m depth using a syringe on the south coast of Curaçao (Water Factory, 12°06′34.18″ N, 68°57′16.22″ W) and the east coast of Brazil (Arraial do Cabo, 22°59′6.96″ S, 41°59′30.57″ W), respectively. One-hundred microliters of the sample was spread onto a marine agar plate (1× marine media in 1.5% agar; BD Difco, USA) and incubated for 24 h at ambient room temperature (31 ± 1°C in Curaçao, 21 ± 0.5°C in Brazil). The resulting mixed community was rinsed off the plate with 3 mL of 0.22 μm-filtered seawater and this stock was diluted to an optical density of 0.50 ± 0.02 AU ($\lambda = 600$ nm).

Sixty microliters of sterile 0.5% top agar (BD Difco, USA) was added to every other well as a control. Then 60 μl of the top agar with the mixed microbial community from *M. annularis* or *M. hispida* were added to the remaining wells. The top agar was kept at 42°C prior to the addition of the mixed microbial community (*Spencer, 1952*). For the microbial wells, the diluted microbial community was added to the warmed top agar in a 1:10 ratio. The resulting plates were allowed to solidify at room temperature and used within 1 h of production.

*Production of exudates:* For experiments involving *M. annularis,* rubble covered with turf or CCA was collected at a depth of <5 m (12°07′18.71″ N, 68°58′09.57″ W) and transported to the lab within one hour. The algae-encrusted rubble was spread over the bottom of a 10 L bucket (380 cm$^2$) and filled with 4 L of raw seawater. The concentration of the resulting exudates was normalized by diluting the resultant exudate water with raw seawater based on the total surface area of the algal substrate used in each preparation. The buckets were kept in diffused sunlight and bubbled with air for a full daylight cycle. The

exudates were filtered through a 10 μm nylon mesh (Filter Specialists Inc., USA) to remove large particulates. The temperature of the exudates was $27 \pm 1°C$ during preparation. For experiments using *M. hispida,* rubble covered with turf algae or CCA was collected at a depth between 5 and 10 m and was kept in a cooler to be transported back to the lab within 4 h. The algae-encrusted rubble was spread over the bottom of 2 L beakers (Pyrex) and filled with 1800 mL of Coral Pro Salt artificial seawater (Red Sea, USA). The beakers were kept under artificial daylight provided by $2 \times 54$ W Marine-Glo T-5 HO (Rolf C. Hagen Inc., Canada) and $1 \times 250$ W 14 K HQI (Hamilton Bulb, USA) bulbs and bubbled with air for a full daylight cycle. The exudates were filtered through 10 μm nylon mesh to remove larger particulates. The temperature of the exudates was maintained at $22 \pm 1°C$. Special care was taken to control for surface cover (area), so the differences that are due to available substrates would likely be seen *in situ* as well.

Incubations were performed with exudates from turf algae, exudates from CCA, or a seawater control. For each exudate treatment, a BOD plate was submerged within one liter of exudate in an acrylic aquarium. Any trapped air bubbles were removed by tapping the wells. The plates were kept in the dark for 7 h and imaged by the SOOpR system every 5 min. Optodes used for these experiments were calibrated at $27 \pm 0.5°C$ and $22 \pm 0.5°C$ for *M. annularis and M. hispida*, respectively.

## Experiment 2 - Single strain studies

One-hundred microliters of the *M. annularis* mucus sample was plated on a 1.5% seawater agar plate (BD Difco, USA) and incubated for 48 h at ambient room temperature ($31 \pm 1°C$). Four colonies with different phenotypes were isolated and streaked onto 1:1 marine media agar plates (BD Difco, USA) and incubated for 24 h at room temperature. The plates of each isolate were rinsed with 3 mL of 0.22 μm-filtered seawater to remove the bacteria, and this stock was diluted to an optical density of $0.5 \pm 0.015$ ($\lambda = 600$ nm). Sixty microliters of sterile 0.5% top agar (BD Difco, USA) was added as a negative control in the columns between each treatment. One of each of the four cultured isolates were added to every other column and incubations were performed in turf algal exudate as described above.

## Experiment 3 - Study of bacterial assemblages associated with different host organisms

Water samples from the surface of turf algae, CCA, and coral were obtained from specimens raised in aquaria at Scripps Institution of Oceanography (San Diego, CA, USA) for ∼6 months. One hundred microliters of each water sample was plated on 1:1 marine media agar plates (BD Difco, USA) and incubated at 30°C for 24 h. The plate was then rinsed with 3 mL of 0.02 μm filtered seawater and the stock was diluted to an optical density of $0.50 \pm 0.02$ AU ($\lambda = 600$ nm). Sixty microliters of sterile 0.5% top agar (BD Difco, USA) was added as a negative control in the rows between each treatment. The remaining rows were filled with the cultured stocks of bacterial communities associated with the different organisms (turf algae, coral and CCA).

Algal exudates were prepared by covering the bottom of a glass aquarium (273 cm²) with turf-encrusted rubble (Scripps Institution of Oceanography, San Diego, USA). The aquarium was then filled with 3 L of raw seawater. Water was bubbled with air for a full daylight cycle and left to exude under artificial light, which was provided by 2 × 54 W 6000 K Aquablue⁺, 1 × 54 W 6000 K Midday, and 1 × 54 W Actinic⁺ aquarium lights (Geismann, Germany). Lights were mounted 80 cm above the experimental area, resulting in photosynthetic active radiation of 120 µmol quanta m⁻² s⁻¹ as measured by a LI-COR LI-193 Spherical Quantum Sensor. The turf exudate was sterilized using a 0.22 µm Sterivex™ filter (Millepore, USA). The temperature of the exudates was 25 ± 1°C. Optodes used for these experiments were calibrated at 25 ± 0.5°C.

*Sequencing of M. annularis isolates:* Genomic DNA was extracted from the four bacterial isolates cultured from *M. annularis* mucus using the NucleoSpin® Genomic DNA Tissue Kit (Macherey-Nagel, Germany). The 16S rRNA genes were amplified by PCR. The bacterial primers 27F (5′-AGAGTTTGATCMTGGCTCAG) and 1492R (5′-TACGGYTACCTTGTTACGACTT) were used and the thermocycler conditions were as follows: initial denaturation step (5 min at 94°C), 30 cycles of denaturation (1 min at 94°C), annealing (30 s at 65°C, −0.5°C per cycle), and elongation (1 min at 72°C), with a final elongation for 10 min at 72°C, then cooled to 4°C. 16S amplicons were verified by visualizing the correct band size using gel electrophoresis (1% agarose gel stained with ethidium bromide). PCR products were then purified using the QIAquick® PCR Purification Kit (Qiagen, The Netherlands). Amplicons were then sequenced in both directions with the forward and reverse primers using Sanger sequencing (Eton Bioscience Inc., CA, USA). Isolates were identified to the nearest known relative using The Ribosomal Database Project (http://rdp.cme.msu.edu).

*Image analysis:* All images were recorded in RAW mode and converted to three 16-bit TIFF images (red, averaged green, and blue) using RawHide image conversion software (v0.88.001, My-Spot Software, USA). The resulting images were imported into ImageJ (v1.44o, NIH Software, USA) to determine red and green pixel intensities for each well. A 300-pixel box was made on the image that filled the area of the well. The average pixel intensity within the region of interest was determined for each well. This process was done for both red and green images. These pixel intensities were used to calculate R, the ratio at a given DO concentration.

$$R = \frac{(\text{intensity of red} - \text{intensity of green})}{\text{intensity of green}} \tag{1}$$

Oxygen concentrations were calculated using the modified Stern-Volmer equation (Eq. (2)), where $R_0$ is the ratio at anoxia, $C$ is the concentration of dissolved oxygen, $K_{sv}$ is the Stern-Volmer quenching constant, and $\alpha$ is the unquenchable fraction of the optode (*Klimant, Meyer & Kuhl, 1995*).

$$\frac{R}{R_0} = \left[ \alpha + (1 - \alpha) \left( \frac{1}{1 + K_{sv} \times C} \right) \right] \tag{2}$$

*Statistical analyses:* Statistics were performed using the software Graphpad Prism and R (Prism version 5, R version 2.13.2). Unless otherwise stated, all statistics were performed on the changes in dissolved oxygen from $T_{initial}$ to $T_{final}$. A Kruskal-Wallis test was used to determine if there were significant differences in dissolved oxygen concentration amongst the treatments ($\alpha = 0.05$). Two-tailed Mann-Whitney tests were performed to determine if the starting concentration of dissolved oxygen in each treatment was statistically different between the control and bacterial wells, as well as between various bacterial treatments. All results are presented as mean $\pm$ standard error (SE).

# RESULTS AND DISCUSSION

## Validation of camera system

A modular camera and excitation system, the SOOpR, which is capable of operating autonomously and to be fully submerged was developed here. Optode excitation analysis in the SOOpR system generated two visible peaks corresponding to Macrolex Yellow (550 nm) and platinum (II) octaethylporphyrin (PtOEP) (650 nm), as expected (Fig. S2). These peaks were consistent with those seen in the study described by *Larsen et al. (2011)*. Additionally, calibration of the oxygen optode sheet was performed to validate whether measured values of DO compared to values calculated from the Stern-Volmer equation. From the recorded image, oxygen concentration of seawater was determined to be 191.6 μM. This was cross-referenced by measuring the oxygen concentration of the same seawater using the LBOD Probe; measured concentration was 188.8 μM. The SOOpR system is capable of being fully submerged to depths greater than 10 m and is able to accurately measure oxygen concentrations. This provides a novel method to estimate bacterial growth rates and oxygen dynamics on the reef and without the use of radioisotopes necessary for other bacterial growth assays (e.g., thymidine and leucine incorporation) (*Fuhrman & Azam, 1982*; *Kirchman, Knees & Hodson, 1985*).

## Bacterial isolates respond differently to turf algal exudate

Individual bacterial isolates cultured from *M. annularis* mucus were tested to investigate whether bacterial strains differentially respond to turf algal-derived exudates. For this purpose, exudates derived from turf algae were used as they proved in previous studies to foster noticeable elevations in microbial oxygen demand (*Haas et al., 2011*). The present study revealed that select coral-associated microbial isolates had significantly different oxygen consumption rates when exposed to the introduced turf exudates. Isolate one consumed significantly more oxygen in the presence of turf exudate than any other isolate (Fig. 2, $84.49 \pm 7.44$ μM, Mann-Whitney test, $p < 0.01$). Sequencing of the 16S ribosomal subunit identified isolates one, two, and four as *Pseudoalteromonas* spp. and isolate three in the family *Bacillaceae*. *Nelson et al. (2013)* found that bacteria from the family *Pseudoalteromonadaceae* were significantly enriched in fleshy algal exudates compared to calcifying organism exudates (i.e., *Halimeda* sp. and coral). This differential response shows that introduction of organic matter sources can differentially alter the activity of microbes naturally residing in the coral mucus.

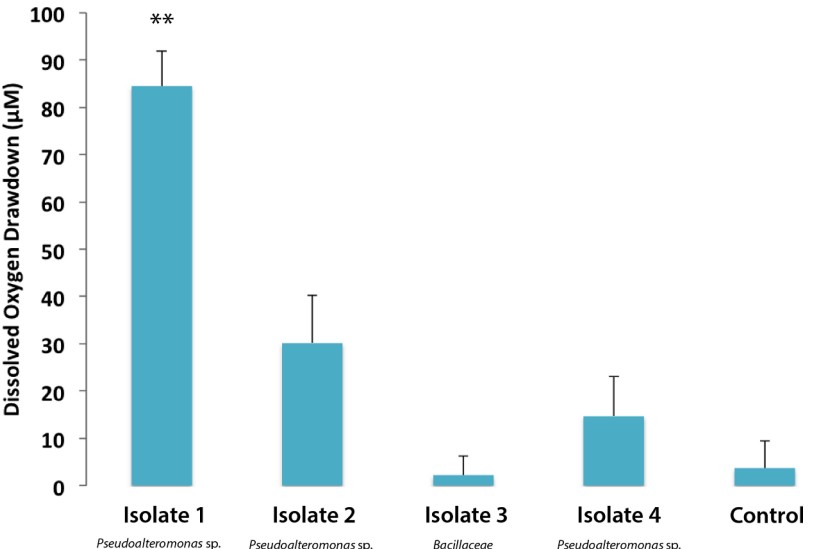

**Figure 2 Dissolved oxygen drawdown by four single bacterial strains isolated from *M. annularis* and exposed to turf exudate.** Oxygen drawdown is shown as the total reduction in oxygen concentration over a 7 h incubation period. $n = 8$ replicates. ** significantly higher oxygen drawdown ($p < 0.01$), $\alpha = 0.05$. Error bars represent + SE.

## Bacterial communities cultured from coral, turf algae, and CCA respond similarly to turf algal exudate

To assess bacterial community responses to algal DOM, bacteria associated with different functional groups (turf/CCA/coral) were exposed to turf algal exudates and the oxygen drawdown was determined. Bacterial communities drew down similar amounts of oxygen when exposed to turf exudate, regardless of their original host (Fig. 3, mean = $304.15 \pm 11.60\,\mu$M). Although individual microbial isolates respond differentially to turf exudate, there appears to be sufficient functional redundancy in all of the communities to respond in a similar manner. However, possible lab culturing bias could account for this observation. These results suggest that it is the addition of turf algal exudates that accounts for the large drawdown of oxygen, as opposed to the specific community of microbes introduced.

## Turf algae elicit the greatest oxygen drawdown by coral-associated bacteria

One mechanism likely contributing to decrease in coral health is hypoxia at borders where corals are in close proximity with turf or macroalgae (*Smith et al., 2006*; *Barott et al., 2009*; *Barott et al., 2012*). Our studies were carried out to investigate how organic matter derived from benthic algae can affect the oxygen consumption rates of microbial communities associated with both corals and algae. To do this, bacterial community cultures from *M. annularis* and *M. hispida* were exposed to exudates from turf algae and CCA to assess whether respiration was differentially affected by algal DOM source. Both *M. annularis and M. hispida* BOD plates showed an overall decrease in dissolved oxygen concentrations in all wells over the period in 7 h (Fig. 4).

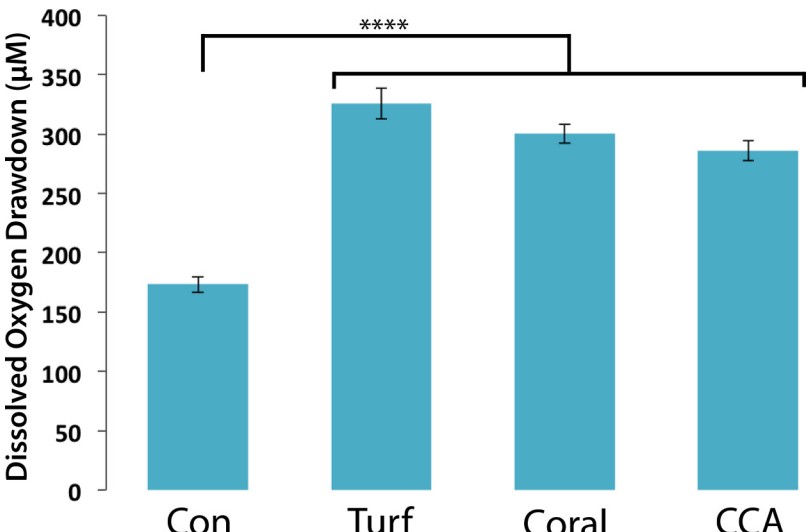

**Figure 3 Dissolved oxygen draw down by bacterial communities cultured from turf, coral and CCA.** All bacterial communities were exposed to 0.22 µm-filtered turf exudate. Microbial communities cultured from turf algae, CCA, and corals were maintained in an aquaria for ∼6 months. No significant difference in oxygen drawdown by different communities of bacteria ($p > 0.05$). **** = $P < 0.0001$ Error bars represent ±SE.

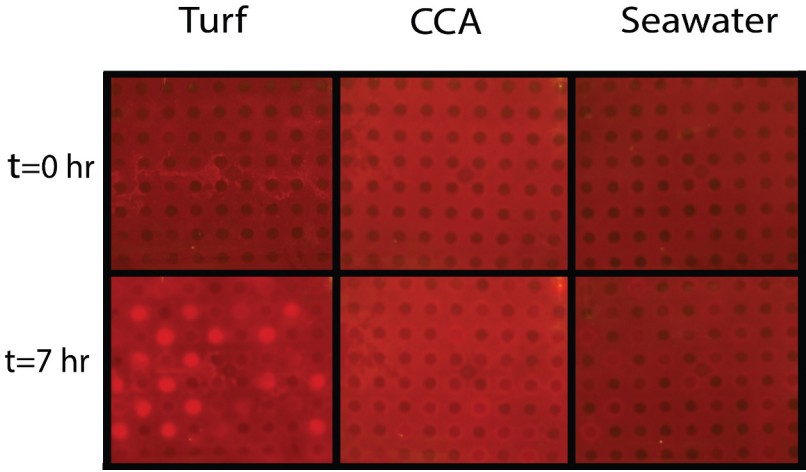

**Figure 4 Experimental biological oxygen demand (BOD) plates.** $T_{initial}$ and $T_{final}$ (after 7 h) images of BOD plates in turf exudate, CCA exudate, and seawater incubations. Bacterial inoculum was distributed in checkerboard pattern starting with the first well containing bacteria. The increase in red luminescent signal is correlated with a decrease in oxygen concentrations.

For the *M. annularis* BOD plates, the DO drawdown of the bacterial wells over 7 h was significantly higher in the turf treatment ($272.45 \pm 11.67$ µM) compared to the CCA treatment ($105.27 \pm 6.79$ µM) and the seawater control ($43.04 \pm 4.88$ µM; Fig. 5A, Mann-Whitney test, $p < 0.0001$). The control wells in the turf and CCA exudate treatment also showed oxygen drawdown over the 7 h period, although not as great as

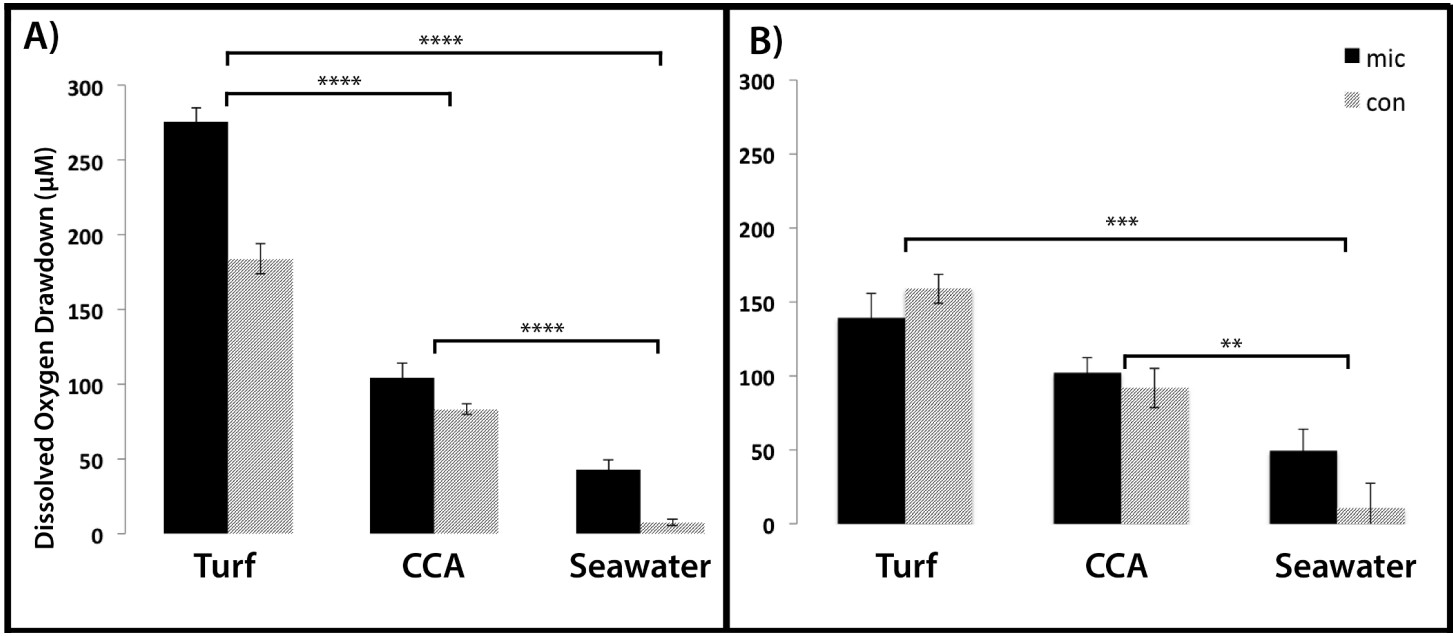

**Figure 5** **Dissolved oxygen drawdown by coral-associated bacteria when exposed to different algal exudate treatments.** Shown is the dissolved oxygen drawdown of microbial communities cultured from (A) *Montastraea annularis* ($n = 30$) and (B) *Mussismilia hispida* ($n = 12$) when exposed to turf algae exudate, crustose coralline algae exudate and seawater for 7 h. Mic and Con correspond to wells with and without bacteria in the agar, respectively. **** = $p < 0.0001$, *** = $p < 0.001$, ** = $p < 0.01$ at $\alpha = 0.05$. Error bars represent $\pm$ SE.

the wells inoculated with bacteria. As with the microbial community associated with *M. annularis*, both exudate treatments caused the microbial communities cultured from *M. hispida* to drawdown significantly more oxygen than the seawater control (Fig. 5B, Mann-Whitney test, $p = 0.0003$ and $0.004$ for turf and CCA, respectively). In the presence of turf algal exudate, *M. hispida*-associated microbial communities also drew down more oxygen as compared to microbial communities exposed to CCA exudate; however, this difference was not statistically significant (Mann-Whitney test, $p = 0.07$). These differences between *M. annularis* and *M. hispida* may be explained by differences in the composition of the microbial communities associated with each of these species (*de Castro et al., 2010*; *Barott et al., 2011*). Similar to the *M. annularis* studies, the controls in the *M. hispida* experiment exposed to both turf and CCA exudate drew down greater than 90 μM of oxygen. There was no statistically significant difference in the DO consumption of the bacterial and control wells within the turf algae (Mann-Whitney test, $p = 0.167$) or CCA (Mann-Whitney test, $p = 0.489$) treatment for *M. hispida*-associated communities. This suggests that the ambient microbes present during the algal exudate incubations (i.e., water column- or algal-associated) were consuming a large amount of oxygen in the presence of this algal-generated organic matter, leading to drawdown of oxygen detected within the control wells (Fig. 5). Together, these data show that bacterial communities are able to consume turf algae exudates more readily than exudates derived from the calcifying CCA. It has been previously shown that turf algae generate more DOC than CCA do, which leads to increased microbial oxygen consumption (*Haas et al., 2011*). In addition,

allelochemicals released by cyanobacteria can stimulate bacterial growth (*Morrow et al.,* *2011*). As cyanobacteria are often present in turf algal assemblages (*Fong & Paul, 2011*), it is possible that these secondary metabolites may have aided in the enhanced microbial growth observed in this study. Further studies to characterize the chemical composition of algal exudates should be conducted.

It has been documented that reef bacteria readily consume algal-derived organic matter (*Cole, Likens & Strayer, 1982*; *Wild et al., 2010*; *Haas et al., 2011*) leading to increased abundances; however, introduction of a novel organic matter source may also alter the community of microbes present (*Carlson et al., 2004*). Naturally occurring coral-associated microbial community structure can shift towards pathogenic bacteria when DOC is introduced (*Thurber et al., 2009*) and when corals are in contact with fleshy algae (*Barott et al., 2012*). Furthermore, pelagic microbial communities incubated with fleshy algal exudates became enriched with copiotrophic bacteria including potential pathogens (*Nelson et al., 2013*). Taken together with these findings, the results of this study indicate that not only can algae-derived dissolved organic matter significantly increase culturable microbial community metabolism, but it can also select for individual strains of culturable bacteria, potentially disturbing the homeostasis of the coral holobiont (*Rohwer et al.,* *2001*; *Rohwer et al., 2002*).

## Caveats

In this study we use nutrient rich media which selects for fast-growing heterotrophic microbes. Although this could confound our results by overestimating the heterotrophic nature of the *in situ* community, *Haas et al. (2011)* showed that turf algal exudate elicited the highest microbial growth response in a culture-independent study of pelagic microbes. Therefore it is likely the trends we observed would also be seen in unculturable microbes. One drawback of our experimental design was that the algal exudates in the field studies were not 0.22 µm-filtered This could possibly confound our results because the microbes that grew during the exuding process could potentially be very different than those naturally residing on the algae due to the effects of growing in a confined space (*Zobell* *& Anderson, 1936*; *Ferguson, Buckley & Palumbo, 1984*; *Lee & Fuhrman, 1991*).

Finally, the exudate concentrations used here likely exceed *in situ* values, which could call into question the biological relevance of the proposed mechanism. However, the accompanying study by *Haas et al. (2013b)* indicates that responses of reef-associated microbial communities are significantly more dependent on the source of exudates than concentration.

## CONCLUSION

This study supports the hypothesis that hypoxia in close proximity to turf algae is driven in part by increased microbial oxygen demand. Despite variation in oxygen demand of individual bacterial strains, oxygen drawdown at the community level showed no significant differences among different community types. Changes in the source of organic matter, on the other hand, affected oxygen consumption rates of cultured bacterial communities significantly. This suggests that hypoxia, commonly found at the interface of

coral-algal interactions, is a result of the complex interactions of coral-associated microbial communities and the properties of the organic matter available. The increased input of bioavailable DOM accompanying phase shifts from calcifier-dominated (i.e., CCA- and coral-dominated) to non-calcifier-dominated (i.e., turf algae- and fleshy macroalgae-dominated) systems can thus change microbial metabolism in a way that oxygen demand outweighs production, possibly leading to a shift towards a net heterotrophic microbial reef community (see companion study *Haas et al., 2013b*). Initially, these effects are likely taking place on a very small spatial scale; however, due to the DDAM positive feedback loop, turf algae are able to slowly crawl further in on the coral eventually overgrowing it, potentially leading to coral-algal phase shifts. Similar results of turf exudate-induced increases of microbial respiration, obtained from two different ocean systems, suggest that this microbially-mediated hypoxic stress could have important implications in the structure and health of the worlds coral reef systems.

## ACKNOWLEDGEMENTS

We thank CARMABI in Curaçao for their facilities and Birch Aquarium at Scripps Institution of Oceanography for supplying us with algal and coral specimens. We thank Franklin Holub for his generous technical support with RawHide and Alex Hewitt for his expertise with CHDK. We thank Linda Kelly and Steven Quistad for their helpful comments throughout the scientific and writing process. We also thank reviewers Kathleen Morrow and Mike Sweet for their extremely helpful comments on this manuscript.

### Funding

Funding for this work was provided by the National Science Foundation to FLR (Grant No. OCE-0927415, OCE-0927448, and DEB-1046413). The funders had no role in study design, data collection and analysis, decision to publish, or preparation of the manuscript.

### Grant Disclosures

The following grant information was disclosed by the authors:
NSF: Grant Nos. OCE-0927415, OCE-0927448, DEB-1046413.

### Competing Interests

Fabiano Thompson is an Academic Editor for PeerJ.

### Author Contributions

- AK Gregg conceived and designed the experiments, performed the experiments, analyzed the data, contributed reagents/materials/analysis tools, wrote the paper.
- M Hatay conceived and designed the experiments, contributed reagents/materials/analysis tools, wrote the paper.
- AF Haas analyzed the data, contributed reagents/materials/analysis tools, wrote the paper.

- NL Robinett performed the experiments, contributed reagents/materials/analysis tools, wrote the paper.
- K Barott, MJA Vermeij, KL Marhaver, P Meirelles and F Thompson contributed reagents/materials/analysis tools, wrote the paper.
- F Rohwer conceived and designed the experiments, analyzed the data, contributed reagents/materials/analysis tools, wrote the paper.

### Supplemental Information

Supplemental information for this article can be found online at http://dx.doi.org/10.7717/peerj.107.

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
