# Peer review of "Biological oxygen demand optode analysis of coral reef-associated microbial communities exposed to algal exudates"

_PeerJ, doi:10.7717/peerj.107_

## Round 0.1 · original submission · Major Revisions

Please pay careful attention to Dr Morrow's comments and suggestions about the experimental design and inferences that can be made from the study. I think I would add some caveats to the Discussion about how these issues (and I know they are complicated to avoid) affect the interpretations.

·

Basic reporting

Although the first paragraph of the introduction is written very nicely, it is largely composed of information that has been reported in multiple recent publications. I’m not sure that there is a need to summarize all of the literature pertaining to detrimental coral-algal interactions, when this paper deals with one aspect – DOC - in relation to turfs and CCA, not foliose macroalgae. I think 1 or 2 sentences pointing to recent reviews on the subject would suffice.

Also, rather than summarize what has been overly summarized already – could you provide some further insight into other systems (non-marine?) where DOC is involved in microbial regulatory mechanisms? If there are any, elaboration here would be quite interesting.

Line 68: Coral recruits preferentially settle on “some types” of CCA – some are inhibitory, so it’s important to not assume all CCA promote settlement.

Figure 2 is difficult to interpret, it needs more explanation. Would it be more informative to say “Dissolved oxygen draw down by four single strains isolated from M. annularis exposed to turf exudates, n=8 replicates. Con = control. * significantly higher oxygen draw down (P<0.01 )” Additionally, to and tf should be defined.

Figure 3 also needs more explanation. These graphs (Fig 2, 3, and 5) might be much easier to interpret if they were displayed as bar graphs of the change in dissolved O2 between tf and t0 with error bars (as in the Haas companion paper). The statistics would be clearer as well. These graphs do present more data as they are now, but it’s not necessary for the interpretation of results. It should also be stated in the legend that these samples were exposed to filtered turf exudate and collected from turf, CCA, and corals maintained in aquaria for XX months. These differences set this experiment apart, quite distinctly, from all the other experiments and it should be clear just from reading the figure legend.

In Figure 5, If you statistically compared the oxygen draw down on the Mic (microbe agar) to the corresponding Con (control agar) – the *** for significance should be between the two, not just over Turf Mic. Again, results would be easier to interpret as bar graphs, unless you think it’s important that the reader knows that the initial oxygen readings varied.

Experimental design

The experimental question is interesting but would have been much less confounded if the turf, CCA, and seawater exudates were 0.22 micron filtered before using with the experimental cultures? After 24 hrs in a bucket, the microbial diversity and abundance in the exudate water would have significantly changed, potentially driving differences between the experimental treatments and masking the actual effect of the coral microbes - rather than the microbes associated with your benthic substrata.

As you mention, this is likely the cause of the oxygen draw down you recorded in the control wells exposed to turf and CCA exudates. Obviously turf algae provide a larger surface area of exposed organic material for bacterial communities to inhabit, in comparison to CCA – thus it is not surprising that your studies would be largely confounded by the diverse microbial communities associated with the turf algae and their exudates – making it difficult or impossible to make any cross-treatment comparisons. I know you tried to get around this by using aquaria reared CCA, Corals, and turf – but it is well known that the microbial communities associated with aquarium organisms are quite different from nature. If experiment had included both filtered and non-filtered exudates, the conclusions would have been more robust for making conclusions about reef wide changes. This was done in the Haas companion paper. Thus, further discussion is needed to explain why you did not filter the exudates and how it might effect your interpretation of results.

Why were the single strain studies only incubated with turf algal exudates and not CCA or seawater? This comparison would have been interesting.

In the organism associated bacteria study, how long were the specimens of turf algae, CCA, and coral living in aquaria? Was a water sample from the inflow processed simultaneously? Why was the exudate filtered in this study and not in the field studies?

Also, it would be beneficial to know the taxonomic identity of the isolates used in this study, particularly since they responded so differently. It is likely that they were mostly Vibrio species, because this genus is most readily cultured from corals. Without knowing their identity, it is hard to say what level of diversity you were able to examine – if all the isolates were Vibrios for example, does the study provide much information about how the hundreds of unique OTU’s associated with corals respond to changes in DOC?

Validity of the findings

Line 315-317. Be careful how much you extrapolate from your data based solely on cultured microorganisms. “Taken together with these findings, the results of this study indicate that not only can algal derived dissolved organic matter significantly increase (INSERT culturable) microbial community metabolism, but can also select for individual strains of (INSERT culturable) bacteria, potentially disturbing the homeostasis of the coral holobiont (Rohwer et al. 2001; 2002).”

This study is dependent on the ability to culture the microbiota in question, which are often heterotrophic in nature, and innately more responsive to DOC sources for nutrition and growth. Be careful not to state more here than you are able to prove with your data.

You artificially enriched a volume of seawater with dissolved organics and microbes to a level that may be difficult or impossible to reach in an open reef environment with reasonable water flow, maybe more plausible with macroalgae than turf, but macroalgae were not examined in this study. Again be careful how much you extrapolate in your conclusions about reef-wide shifts in microbial metabolism and their effect on the entire coral colony (holobiont). It is alright to speculate, but make sure it is identified as such.

Additional comments

This was an interesting study, it just needs to be cleaned up a bit. I apologize if any of my comments are unclear, but I'm happy to answer any additional questions you have about the review. Feel free to email me at [email protected] and good luck with your revisions.

·

Basic reporting

The article is very well written overall and should be accepted for publication

Experimental design

The methodology is clear and concise, my only point would be the reference to one paper in prep and one submitted as the readers have no way to see these, i'm would suggest ommiting these and adding in any extra detail needed into this paper.

Validity of the findings

very interesting and useful to coral science, the development of new protocols/equipment will always advance any particular field

Additional comments

The paper is well written, I only have some very minor corrections to suggest to the authors

Introduction
Line 51 Halimedia in italics
Line 55-56 sort out references
Line 73 rearrange brackets or have double bracket at end of ref
Line 78 ref Hass et al
Line 115 25 what?
Line 118 °C
Line 215 insert ‘to’ before determine
Line 235 not ideal to reference in prep work in my opinion can this be adjusted some how
Line 292 space after algae

---

## Round 0.2 · Minor Revisions

As the authors may know, I have long been skeptical of the broad significance of the "DDAM" model. My group has even published a paper that describes a field test of it; Vu et al doi:10.1371/journal.pone.0004514 that unsurprisingly never seems to get cited by the DDAM proponents. Regardless, I found this to be an interesting and important paper describing a careful and novel experimental approach testing how the exudates of turf and CCA affect benthic metabolism (in a sense). I found the paper to be well written and the inferences to be careful and restrained, e.g., I appreciate the clear caveats at the end of the Results and Discussion section and the use of retrained language when speculating on the implications of the study.

I think you have done a good job in responding to the reviewer comments and in modifying the paper accordingly (mainly adding caveats); thank you. I have five more minor changes to suggest:

1) Reword the first sentence of the Introduction:

line 43: "Anthropogenic inputs such as over-fishing" I would not refer to fishing as an "input". Id rewrite this phrase to just say "Over-fishing and eutrophication are associated with increasing abundances of turf algae"

However: Certainly fishing and nutrient pollution are causally related to macroalgal cover/biomass, but turf? I know some people think this, but what science actually demonstrates it? Let's look at the citations:

McClanahan et al. 1999 and McClanahan et al. 2002 are not in the Lit Cited and since Tim publishes about a hundred papers a year, it is hard to know for sure what is being cited for what. I'm guessing the 2002 cite is for their Belize experiment, which I agree, does indicate that experimental nutrient pollution can promote turf.

Hughes et al. 2007 excluded all fishes and many macoinvertebrates, so I don't agree with the authors (of Hughes et al or Gregg at al) that this is a reasonable test of the effects of fishing on algae or reefs. Also, their cage exclusion treatments caused a decrease, not an increase in turf; "In response to the experimental exclusion of larger herbivorous fishes, benthic assemblages in the cages followed a fundamentally different trajectory over time, with upright fleshy macroalgae rather than corals and algal turfs becoming predominant, mimicking similar responses on many overfished and polluted reefs worldwide [4–8, 20–21]." Hughes et al also imply the pattern is the opposite of what Gregg et al suggest: Hughes et argue here and elsewhere that fishing and pollution promote macroalgae thereby reducing turfs.

Francini et al 2013 (and note it is Francini-Filho) also found a positive association between turf and corals and more turf in the no-take reserves that were sampled.

Sandin et al. 2008 sampled four sites, and did find less turf where there were more fish, but I suspect that had more to do with much lower coral cover but who knows. There is a lot going on there and it is a very small sample size.

Four more minor comments / questions:

From the Abstract:

"Algal-derived dissolved organic matter HAS BEEN HYPOTHESIZED TO induce mortality of reef building corals"

"This suggests that the driving factor for elevated oxygen consumption rates is the source of exudates" Or is it exudate concentration? i.e., rather than composition? (i.e., because concentration is related to the source.)

Discussion:

"Various studies have demonstrated that dissolved oxygen (DO) concentrations are significantly reduced at coral-algal interfaces (Barott et al. 2009, 2011b) and in the water column overlying algal beds (Haas et al. 2010, Wild et al. 2010)"

I don't follow this microbe literature, but I understood that there were some papers that found otherwise - are you citing all relevant literature, including contradictory literature? Also, during the daytime, the presence of macroalgae really reduced benthic DO? That surprises me but if true is pretty remarkable. Again, please be careful to avoid selective citation - is there other relevant work?

Figures: What are error bars in Figs 2, 3, and 5? (add into to Fig legends)

If you can address these 5 questions / suggestions, and revise the manuscript, I will respond very quickly once I get it. In fact, I apologize for the delay in reviewing this revision.

---

## Round 0.3 · accepted · Accept

Thank you for being so fast and efficient in revising the manuscipt.